# Stereotactic Body Radiation Therapy (SBRT) for Renal Cell Carcinoma with Inferior Vena Cava Tumor Thrombus

**DOI:** 10.3390/cancers17040710

**Published:** 2025-02-19

**Authors:** Ilona Pilosov Solomon, Gal Rinott Mizrahi, Ilan Klein, Yoram Dekel, Yuval Freifeld

**Affiliations:** Carmel Medical Center, Haifa 3436212, Israel

**Keywords:** stereotactic body radiation therapy, renal cell carcinoma, inferior vena cava tumor thrombus

## Abstract

Renal cell carcinoma (RCC) is a type of kidney cancer that can spread into the veins, sometimes reaching the large vein (inferior vena cava, IVC) and causing severe health complications. Treatment for this condition typically involves surgery, which can be risky, especially for elderly or frail patients. Recent studies suggest that stereotactic body radiation therapy (SBRT), a method that delivers precise, high doses of radiation to the tumor, could be an alternative for patients who are not suitable for surgery. This review explores the potential of SBRT as a treatment for RCC with IVC involvement (tumor thrombus), focusing on its safety, effectiveness, and ability to reduce symptoms in patients who cannot undergo surgery. Findings indicate that SBRT may offer a promising solution for managing these patients, providing local control of the tumor and reducing associated complications, although more research is needed to confirm long-term benefits.

## 1. Introduction

Renal cell carcinoma (RCC) represents 3–5% of all adult malignancies. The incidence of RCC has been increasing in recent decades, with the highest rates observed in developed countries [1]. Worldwide, there are more than 140,000 RCC-related deaths each year, and RCC ranks as the 13th most common cause of cancer death [1].

RCC intravascular tumor extension, also known as tumor thrombus (TT), occurs in 4–36% of cases and may involve the renal vein, inferior vena cava, and even extend to the right atrium [2].

The presence and location of TT dictate tumor staging, patient prognosis, and management [2,3]. Up to 10% of RCC patients present with inferior vena cava TT (IVC TT) [4]. In the absence of distant metastasis, the standard of care for IVC TT management is surgical extirpation with radical nephrectomy and IVC-TT thrombectomy [3,4,5]. However, this is a complicated procedure, sometimes requiring the use of cardiopulmonary, with a complication rate as high as 35% and up to 10% perioperative mortality rate [6], rendering it unsuitable for elderly and frail patients [2,5].

If IVC TT is left untreated, it may locally progress and metastasize [5] leading to severe, life-threatening sequelae such as Budd–Chiari syndrome and pulmonary embolism.

RCC was traditionally considered to be a radioresistant tumor. In recent years, multiple reports have demonstrated its sensitivity to stereotactic body radiation therapy (SBRT), first, in the metastatic setting, and later, for locally advanced disease and IVC TT, either as a solitary treatment for those unfit for major surgery or in the neoadjuvant setting [2,5,7].

Herein, we will review the major aspects of SBRT use for RCC with IVC TT.

## 2. Methods

We conducted a PubMed and Scopus search for manuscripts in English published between 2000 and 2024. Keywords included the following combinations: “RCC and SBRT” or “kidney cancer and SBRT” or “RCC and stereotactic ablative body radiation (SABR)” or “kidney cancer and SABR”. After duplicated articles were removed, titles and abstracts were screened for the term “IVC”. The article selection process is described in Figure 1.

## 3. SBRT Mechanism

SBRT is a radiotherapy modality that delivers high doses of radiation to well-defined targets with high accuracy, in a limited number of fractions [4]. This approach allows for ablative doses of radiation to be delivered to the tumor while relatively sparing the nearby organs. 

RCC has long been considered resistant to radiotherapy using conventional fractionated doses, based mainly on early studies evaluating the response of RCC cell lines to radiotherapy, which showed relatively poor results [8]. However, emerging evidence suggests that RCC is sensitive to extremely hypo-fractionated radiation such as SBRT [9].

In vitro models published in the late 1990s were the first to suggest the radiosensitivity of RCC cell cultures to extremely hypo-fractionated radiotherapy [10]. Further work in mice models with implanted RCC cells supported these findings. For example, in a study by Walsh et al., the treatment of human RCC in a mouse model with 48 Gy divided into three fractions resulted in a sustained decrease in tumor size [11].

Unlike conventional radiation therapy, which causes DNA damage and increases hypoxic burden that may induce angiogenesis in RCC, thus making it “radioresistant”, SBRT causes both direct and indirect tumor cell death [12,13]. First, SBRT directly induces necrotic cell death. The indirect effect is attributed to rapid endothelial cell apoptosis, thereby damaging the tumor microenvironment [12,13]. Furthermore, as a result of SBRT-induced tumor necrosis, there is a massive release of antigens, which may stimulate an anti-tumor immune response, even outside the irradiated field [13,14].

## 4. The Role of SBRT in RCC Management

### 4.1. SBRT for Local RCC

The gold standard for localized RCC treatment is partial or radical nephrectomy. However, a significant proportion of newly diagnosed patients are over 70 years old [7]. Advanced age correlates with a higher prevalence of comorbidities and reduced renal function, making some of these patients frail and unsuitable for surgery. In such cases, smaller tumors (<4 cm) are frequently managed conservatively with surveillance. It is worth noting that up to 40% of these tumors may exhibit growth and require definitive treatment. Conversely, larger diagnosed tumors typically necessitate immediate treatment [7].

In this context, one treatment option, whether applied initially or after a period of surveillance, is thermal tumor ablation (TA). However, TA is associated with a lower rate of local control and a higher risk of recurrence compared to surgery, particularly in cT1b tumors. Additionally, there are concerns regarding the treatment of perihilar or centrally located tumors, which can pose technical challenges during ablation and may lead to significant complications such as bleeding or injury to the collecting system, potentially requiring radical nephrectomy.

SBRT for localized RCC was initially described in the early 2000s [15]. SBRT has demonstrated high local control rates, reaching 97% at 28 months, in treating clinically inoperable patients with localized cT1a and cT1b tumors, with a favorable toxicity profile [7].

Recently, the FASTRACK II trial, a prospective non-randomized phase 2 study, investigated the efficacy of SBRT in patients with primary RCC who were medically inoperable or unsuitable for surgery. Conducted across multiple centers in Australia and the Netherlands, the trial enrolled patients with biopsy-confirmed primary RCC who had a single lesion and an Eastern Cooperative Oncology Group performance status of 0–2. Patients received either single-fraction SBRT (26 Gy for tumors ≤ 4 cm) or multi-fraction SBRT (42 Gy in three fractions for tumors of 4–10 cm). The primary endpoint was local control, defined as no progression of the primary RCC. Results showed that SBRT provided a 100% local control rate at 12 months, with an acceptable side-effect profile. Seven patients (10%) experienced grade 3 adverse events, but there were no grade 4 events or cancer-related deaths. The study concluded that SBRT is a promising alternative for patients with primary RCC, particularly those inoperable or at high surgical risk. The trial results support the design of future randomized studies comparing SBRT to surgery for localized RCC [16].

In addition, the IROCK consortium reported the largest series to date affirming the long-term efficacy of SBRT in localized RCC. SBRT was administered either as a single fraction (81 patients) or multiple fractions (109 patients), with a median tumor diameter of 4 cm. The cumulative incidence of local failure over 5 years was 5.5%, alongside impressive cancer-specific survival (CSS) and progression-free survival (PFS) rates of 92% and 63%, respectively. Subgroup analysis based on tumor size and fractionation revealed that patients receiving single-fraction SBRT for tumors smaller than 4 cm achieved the best PFS and CSS rates, with the lowest rates of local and distant failures [17].

Another potential advantage of SBRT for localized RCC is the preservation of renal function, as it has minimal impact on the glomerular filtration rate [18].

Regardless of these encouraging data, it is important to note that most findings are based on retrospective series with varying inclusion criteria, as well as non-uniform SBRT protocols, follow-up protocols, and definitions for local disease control. Additionally, up to 15% of patients exhibited progression to metastatic disease, a risk that needs to be considered [17].

### 4.2. SBRT for Metastatic RCC

SBRT has long been utilized for patients with brain and bone metastases from RCC. A meta-analysis by Zaorsky et al. demonstrated that in oligometastatic RCC, with both central nervous system (CNS) and extra-CNS metastases, SBRT was effective and safe, achieving a 90% efficacy rate at one-year follow-up, with only 1% experiencing grade 3 or higher toxicity [19]. In oligometastatic patients naïve to systemic treatment, SBRT has shown a 1-year PFS rate of 82% and an OS rate of 95%, without significant impact on quality of life [20].

## 5. RCC with Venous Invasion

A total of 5 to 10% of RCC patients are diagnosed with tumor extension into the venous system, encompassing the renal vein, inferior vena cava (IVC), and occasionally the right atrium [21]. In over 95% of these cases, patients present with symptoms such as hematuria, flank or abdominal pain, and constitutional symptoms [22]. Clinical signs may include a palpable flank or abdominal mass, lower extremity edema, right-sided varicocele in men, dilated superficial abdominal wall veins, Budd–Chiari syndrome, and pulmonary embolism. These patients require curative and often palliative treatment.

### 5.1. Standard Treatment

Primary localized RCC with IVC TT is typically managed with radical nephrectomy and thrombectomy. The choice of surgical approach depends largely on the level of tumor thrombus, classified by its extension into three anatomical landmarks: the renal vein, sub-hepatic veins, and diaphragm [22,23]. Median OS following extirpative surgery ranges from 38 to 116 months, a stark contrast to the median survival of only 5 months for untreated patients [24]. However, surgery poses significant challenges, particularly in cases of level III tumor thrombus (involves the intrahepatic IVC) and level IV tumor thrombus (extends above the diaphragm with possible extension into the right atrium), often necessitating a multidisciplinary approach and potentially cardiopulmonary bypass. This procedure carries a considerable risk of perioperative morbidity (up to 35%) and mortality (up to 13%) [4,25].

Given that not all patients are suitable candidates for surgery, preliminary data suggest that SBRT may complement conventional treatments for achieving local disease control, either independently or in conjunction with systemic therapy.

### 5.2. SBRT for IVC TT Management

SBRT was initially utilized for treating portal vein and IVC tumor thrombi in hepatocellular carcinoma [26,27]. It has demonstrated promising response rates, achieving complete response in 36% of patients and partial response in up to 78% of patients with IVC TT. These findings have encouraged researchers to explore the application of SBRT for managing RCC with IVC TT (Table 1).

Several case reports have been published in this context. Hannan et al. documented their experience with two patients who underwent SBRT for the treatment of IVC TT. One patient was deemed unfit for surgery, while the other had recurrent IVC TT following prior radical nephrectomy and IVC thrombectomy. SBRT was well tolerated, with no reported early or late toxicities [2]. Subsequent follow-ups of these patients revealed OS ranging from 18 to 36 months without additional systemic treatment [28].

Marchand et al. reported a case involving recurrent IVC TT alongside metastatic disease. The patient received treatment consisting of 5 fractions of 7 Gy to the IVC TT, followed by nivolumab monotherapy. The only early adverse event reported was transient grade 2 nausea, with no late toxicities noted. Subsequently, the patient underwent SBRT directed at a solitary lung nodule. As of the latest follow-up at 54 months, the patient remained alive with no evidence of IVC TT or metastatic disease [29].

The largest multicenter case series, published by Freifeld et al., involved 15 RCC patients with IVC TT treated with SBRT across six leading international centers [30]. More than 50% of the patients had high-level IVC TT, and over 60% had metastatic disease. The median SBRT dose administered was 40 Gy, delivered over a median of five fractions. A significant portion of patients did not undergo surgery due to high surgical risk (46%), recurrent thrombus following prior radical nephrectomy and IVC thrombectomy (20%), and the presence of inoperable tumors and disseminated metastatic disease. Five patients underwent SBRT for palliation of symptoms related to IVC TT, which included Budd–Chiari syndrome, leg edema, hematuria, and pain. All five patients reported symptomatic relief, indicating the effectiveness of SBRT in managing symptoms associated with IVC TT in the palliative care setting. Among patients with post-SBRT imaging available, nearly 60% showed thrombus regression, while 25% had stable disease and only 16% experienced local tumor progression. These findings underscore the potential efficacy of SBRT for treating IVC TT and further support the observed symptomatic improvements mentioned earlier. Median OS was 34 months, with a median survival of 18 months in patients who had no additional systemic therapy. Short-term adverse events included nausea, fatigue, and dermatitis, all classified as minor (grades 1–2) [30].

All the aforementioned manuscripts are limited by being case reports or retrospective case series; however, when taken together, these data suggest the safety and potential efficacy of SBRT as a treatment for RCC with IVC TT. This therapeutic option may offer several advantages, including alleviating associated tumor-related side effects with minimal treatment-related adverse events. It may also postpone or mitigate the need for systemic therapy and potentially serve as definitive therapy in certain cases.

**Table 1 cancers-17-00710-t001:** Studies of SBRT for RCC IVC-TT management.

Author, Year of Publication	Patients (N)	Study Design	Reasons for SBRT to IVC-TT	Metastasis (N of Patients)	IVC-TT Level (Level—N of Patients)	Follow-Up (Months)	Previous Treatments	Total Dose	Number of Fractions (Median)	Survival (Months)	Toxicity
Hannan, 2015 [2] * Freifeld, 2019 [28]	2	Case report	Non-surgical candidates	1	IV—2	18–34	Previous nephrectomy and thrombectomy—1	20–45 Gy	4–5	18–34 (OS)	None
Margulis, 2021 [31]	6	Prospective phases 1 and 2	Neoadjuvant therapy	3	I—2 II—4	24	Non	40 Gy	5	24 (all patients survived throughout follow-up)	Grades 1–3
Freifeld, 2022 [30]	15	Retrospective	Non-surgical candidates	10	I—2 II—5 III—4 IV—4	19	Previous nephrectomy and thrombectomy—4 Previous systemic treatment—11	40 Gy	5	34 (median OS)	Grades 1–2
Castelnau-Marchand, 2023 [29]	1	Case report	Non-surgical candidate	1	IV—1	48	Previous nephrectomy and thrombectomy	35 Gy	5	48 (patient survived throughout follow-up)	Grade 2
Chen, 2024 [32]	8	Prospective	Neoadjuvant therapy	1	II—4 III—4	26	Non	30 Gy	5	30.67 ± 5.38 (OS)	Grades 1–2

N—number, SBRT—stereotactic body radiotherapy, RCC—renal cell carcinoma, IVC-TT—inferior vena cava tumor thrombus, OS—overall survival. * This case report and the one by Freifeld et al. (2019) [28] include the same patients, and both are included in the retrospective series by Freifeld et al. (2022) [30].

### 5.3. SBRT as Neoadjuvant Treatment

Initial investigations are underway for the use of SBRT to IVC TT in the neoadjuvant setting. The rationale for this approach relies on the theory that microemboli released by tumor manipulation during surgery may increase the risk of subsequent metastatic disease. Another possible mechanism for recurrent disease following surgery may be microinvasion into the minor hepatic veins, which cannot be observed or controlled properly during surgery, and similarly, microinvasion into the IVC wall. By radiating the IVC-TT before surgery, it is thought that both of these mechanisms might be mitigated.

Initial safety results of a phase II clinical trial by Margulis et al. reported the outcomes of six patients treated with neoadjuvant IVC TT SBRT, followed by radical nephrectomy and IVC thrombectomy [31]. After a median follow-up of 24 months, all patients were alive. No grade 4–5 adverse events were observed, and it was concluded that neoadjuvant SBRT was a feasible and safe treatment option. The study is about to finish accruing and final results are expected. Although larger-scale prospective phase III randomized controlled trials are needed, these are unlikely in the near future due to the relative rarity of this condition [32].

Chen et al. recently reported a similar study involving eight patients receiving neoadjuvant SBRT [32]. A 30 Gy dose in five fractions was administered, followed by extirpative surgery. The primary endpoint was defined as safety. Echoing the report by Margulis et al. [31], only low-grade adverse events were reported, including anemia, nausea, vomiting, fatigue, and surgery-associated complications. It is important to note that the authors did not provide a clear list of complications or specify whether they were associated with surgery or SBRT. However, as mentioned, no high-grade complications were reported. In this study, repeat imaging prior to surgery showed only minor changes to IVC-TT dimensions. This is supported by the report by Freifeld et al., which indicated a relatively prolonged time to best radiographic response of 478 days, with only two patients achieving their best response within 90 days of completing SBRT [30].

These limited data support the feasibility and rationale of neoadjuvant SBRT. However, a few points should be considered. Based on our experience and the aforementioned studies, SBRT should be administered relatively close to surgery to minimize risks associated with operating on a radiated field and complications from severe fibrosis and local reactions. This emphasizes that neoadjuvant SBRT is not intended to reduce thrombus size or facilitate a technically simpler surgery. Therefore, the primary indication remains local and distant disease control, which should be demonstrated by prolonged DSS and OS in future studies.

## 6. Ongoing Trials and Future Directions

Due to the relative rarity of this condition, large prospective randomized trials are not likely to emerge in the near future. To our knowledge, there is currently one ongoing prospective trial (NCT02473536) aimed at evaluating the safety and efficacy of neo-adjuvant SBRT followed by open nephrectomy and thrombectomy. This is a single-arm, prospective, non-randomized trial including patients with Mayo level I-IV IVC-TT planned for extirpative surgery. All patients will receive neoadjuvant SABR (40 Gy in 5 fractions) aimed at the IVC-TT only within 30 days prior to surgery. Such a treatment plan aims to maximize the effect and benefit of SBRT while avoiding local fibrosis and tissue response which occurs later. So far, promising initial safety results have been published [31] and accrual is ongoing.

Future perspectives regarding SBRT for IVC-TT may include:

Prospective trials with unified protocols and inclusion criteria as the sole treatment for inoperable patients or in combination with systemic treatment.

Evaluating the immune response following SBRT to IVC-TT.

Evaluating the radiological response following SBRT to IVC-TT.

## 7. Discussion

Although limited data exist, large-scale high-level evidence is unlikely to be available due to the rarity of this condition. Taken together, these data suggest the efficacy of SBRT for IVC-TT in several settings:

First, for those unfit for surgery or facing complications with recurrent thrombi, SBRT may offer good local control rates, with stable or regression of thrombi size observed in the majority of cases. While this treatment may not provide a cure, it can alleviate symptoms associated with progressive thrombi. In this context, there is cautious optimism regarding an oncological benefit associated with SBRT, supported by multiple reports indicating relatively prolonged overall survival (OS) compared to the existing literature on untreated IVC-TT [33]. Despite the potential contribution of novel immunotherapy regimens to OS [34], several reports suggest that even without systemic therapy following SBRT, a favorable prognosis may be expected.

Second, for local symptom palliation or for patients with metastatic disease receiving systemic treatment who have symptomatic IVC-T, SBRT may offer an efficient palliative method with relatively minor associated complications.

Finally, SBRT appears to be safe and feasible in the neoadjuvant setting. This option does not appear to be associated with technical difficulties during surgery or increased postoperative complications. However, the oncologic benefit of this approach is yet to be established.

## 8. Conclusions

SBRT is an emerging treatment option for selected RCC patients with IVC TT who are unfit or unwilling to undergo extirpative surgery. Further data are needed to establish the efficacy and safety of this treatment option.

## Figures and Tables

**Figure 1 cancers-17-00710-f001:**
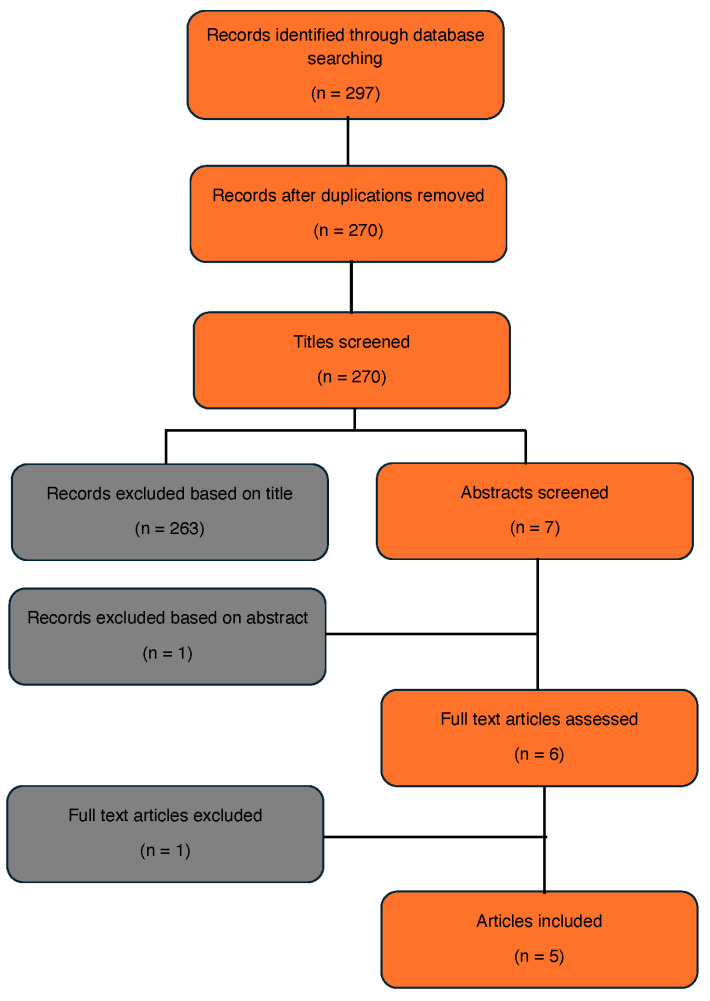
Article selection process.

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
