# Peer review of "Stereotactic Body Radiation Therapy (SBRT) for Renal Cell Carcinoma with Inferior Vena Cava Tumor Thrombus"

_cancers, 2025, doi:10.3390/cancers17040710_

Round 1
Reviewer 1 Report (Previous Reviewer 2)
Comments and Suggestions for Authors
This paper reviews the use of Stereotactic Body Radiation Therapy (SBRT) in treating Renal Cell Carcinoma (RCC) with Inferior Vena Cava Tumor Thrombus (IVC TT). It systematically examines existing clinical data and discusses the mechanisms of SBRT and its potential in RCC treatment, especially for patients who are unfit for surgery. This paper provides a solid background on SBRT's role in both localized and metastatic RCC treatment with up-to-date research findings and highlights the advantages of SBRT, such as high precision and relatively low side effects. As a revised version, this article has made partial improvements to the previous version and is now worthy of acceptance.
Author Response
Comment:
This paper reviews the use of Stereotactic Body Radiation Therapy (SBRT) in treating Renal Cell Carcinoma (RCC) with Inferior Vena Cava Tumor Thrombus (IVC TT). It systematically examines existing clinical data and discusses the mechanisms of SBRT and its potential in RCC treatment, especially for patients who are unfit for surgery. This paper provides a solid background on SBRT's role in both localized and metastatic RCC treatment with up-to-date research findings and highlights the advantages of SBRT, such as high precision and relatively low side effects. As a revised version, this article has made partial improvements to the previous version and is now worthy of acceptance.
Response: Thank you for reviewing the manuscript, we appreciate your time and feedback.
Reviewer 2 Report (Previous Reviewer 1)
Comments and Suggestions for Authors
I would like to thank the authors for the effort of providing a revised version of their manuscript. I would like to add a few clarifications (minor corrections):
1. I can't see Figure 1 and Table 1 in the present version
2. In Methods, the sources of information have been detailed but it is necessary to describe how they have been filtered, analyzed and summarized.
3. In the section "Mechanisms of SBRT" a reference is required for the phrase "Unlike conventional radiotherapy...". It is necessary to justify the claim that conventional RT causes hypoxia and stimulates angiogenesis.
4. In section 4.1, a reference to the FASTRACK study (DOI: 10.1016/S1470-2045(24)00020-2) should be included and the statement "all findings are based on retrospective studies" should be modified
5. It would be useful to explain to the reader the meaning of level III-IV IVC TT (section 5.1)
Author Response
Dear reviewer,
Thank you for the valuable feedback, we appreciate your time spent to improve the quality of our manuscript.
Comment 1: I can't see Figure 1 and Table 1 in the present version
Response 1: We uploaded them once again in the proper section (the table is in the supplementary section, the figure in the figure section
Comment 2: In Methods, the sources of information have been detailed but it is necessary to describe how they have been filtered, analyzed and summarized.
Response 2: Thank you for the comment, after conducting a PubMed and Scopus search with the specified keywords, we first removed any duplicate articles. Then, we screened the titles and abstracts for relevance, specifically looking for the term "IVC" to identify studies that were pertinent to our review of RCC with IVC tumor thrombus treated with SBRT. Articles that met these criteria were selected for full-text review.
The entire selection process, including the steps for inclusion and exclusion, is visually depicted in Figure 1. We encourage you to refer to this figure for a clear representation of how articles were filtered and analyzed. Additionally, to ensure comprehensiveness, we performed a secondary search using the Research Rabbit AI engine to identify articles similar to references 28-30. We hope this explanation addresses your concern, and thank you again for your helpful feedback.
Comment 3: In the section "Mechanisms of SBRT" a reference is required for the phrase "Unlike conventional radiotherapy...". It is necessary to justify the claim that conventional RT causes hypoxia and stimulates angiogenesis.
Response 3: We have now added a reference in that section.
Comment 4: In section 4.1, a reference to the FASTRACK study (DOI: 10.1016/S1470-2045(24)00020-2) should be included and the statement "all findings are based on retrospective studies" should be modified
Response 4: Thank you for your comment. We have incorporated the findings of the FASTRACK II trial into the manuscript, which provides important prospective data on the efficacy of SBRT for primary renal cell carcinoma.
Comment 5: It would be useful to explain to the reader the meaning of level III-IV IVC TT (section 5.1)
Response 5: Thank you for your helpful suggestion. In response, I have added a brief explanation of level III-IV IVC tumor thrombus (IVC TT) in Section 5.1.
This manuscript is a resubmission of an earlier submission. The following is a list of the peer review reports and author responses from that submission.
Round 1
Reviewer 1 Report
Comments and Suggestions for Authors
This manuscript addresses a timely topic and its importance is explicitly justified. However, some major revisions are needed before it can be published
Aims are formulated generally but not concretely in terms of clear questions. Specific endpoints should be formulated for curative and palliative treatments.
The scope of the review is broader than the initial author's declaration (title and introduction). The review includes not only locally advanced tumors but the radiobiological mechanisms of SBRT, and SBRT reports on localized and metastatic RCC. The reviewed references should address specifically the declared article's purpose.
The literature search is described briefly. No selection criteria were used.
Key statements are generaly supported by references with only some exceptions: abscopal effect is not reported in the series on table 1. Therefore, the final paragraph in the discussion should be removed.
Appropriate evidence is generally present but data could be presented in a more appropriate way. For instance, Table 1 should indicate patient's and tumor's characteristics in more detail (metastatic, palliative, thrombus level, previous treatments, toxicity rates).
Minor:
In the introduction: cardiopulmonary ¿bypass?
Reviewer 2 Report
Comments and Suggestions for Authors
This article reviews the existing evidence on the use of SBRT in patients with RCC combined with IVC TT and considers its application as a treatment, palliative, and neoadjuvant therapy. However, the review is too simple, as the authors listed previous research simply. Maybe, the authors should show more details about the previous research, and elaborate more on their views on these studies.